# Responses of microbial community dynamics, co-occurrences, functional shifts, and natural fermentation profiles of *Elymus nutans* silage to altitudinal gradients

Rina Su,[1,2] Fuhou Li,[1,2] Ying Liang,[1,2] Neha Sheoran,[1,2] Jie Bai,[3] Lizhuang Hao,[4] Wencan Ke,[5] Chen Hu,[1,2] Mengya Jia,[1,2] Samaila Usman,[1,2] Mengyan Chen,[1,2] Xusheng Guo[1,2]

**ABSTRACT**   This study investigated the microbial diversity, functionality, and fermentation quality of *Elymus nutans* (*E. nutans*) silage across different altitudinal gradients. Forage samples were gathered from three areas in Qinghai province, China, namely, Huangyuan (HY, altitude of 2,613 m); Xinghai (XH, altitude of 3,612 m), and Chenduo (CD, altitude of 4,589 m) counties. Chopped *E. nutans* were ensiled at local room temperature and sampled after 7, 14, 30, and 60 days of ensiling. The results exhibited that the bacterial diversity and numbers of the lactic acid bacteria (LAB) and yeast in *E. nutans* forage decreased, while the water-soluble carbohydrates and crude protein increased along the altitudinal gradients. This led to clear differences in fermentation qualities and bacterial community dynamics. The silage from CD with high altitude showed a higher fermentation quality, more stable bacterial network structure, and lower abundance of amino acid metabolism than ensiled forages collected from the other regions (*P* < 0.05). Meanwhile, the LAB including *Lacticaseibacillus* and *Levilactobacillus* were identified as differentially abundant taxa in silage sample from CD after 60 days of ensiling; in contrast, the undesirable microbiota were still the differentially abundant taxa in the silages from HY and XH. Correlation analysis further confirmed that altitude affects the epiphytic microbial structure and nutrition distribution of *E. nutans* forage, leading to varying levels of fermentation. Considering the complex microbial composition of fresh forage, LAB inoculation is highly suggested for *E. nutans* silage making in the Qinghai regions with low altitudes.

**IMPORTANCE** On the Qinghai-Tibet Plateau (QTP), feed shortages are common due to cold environmental conditions and the short growing season of crops. Therefore, effective preservation, such as the ensiling of local forage, is becoming increasingly important to balance the seasonal imbalance between the forage supply and the nutritional needs of domestic animals in this area. However, the structure of the microbial community of the forage, which is influenced by climatic conditions such as altitude differences, has a major impact on the fermentation quality and microbial succession of the ensiled forage. Therefore, we investigated microbial community dynamics, co-occurrence, functional shifts, and natural fermentation profiles of *Elymus nutans* silage as a function of altitudinal gradients. Results show that silage from Chenduo at higher elevations has better fermentation quality and higher abundance of *Lacticaseibacillus* and *Levilactobacillus* than ensiled forage from other regions. This work may contribute to guiding for silage production in QTP.

**KEYWORDS**   *Elymus nutans*, fermentation quality, bacterial community, altitudinal gradients, Qinghai region

Address correspondence to Xusheng Guo, guoxsh07@lzu.edu.cn.

The authors declare no conflict of interest.

See the funding table on p. 14.

The Qinghai-Tibet Plateau (QTP) is one of the most important alpine husbandry areas in China (1). However, due to the frigid environmental conditions (e.g., low temperature, strong ultraviolet radiation, and hypoxia) and the short growing season of plants in this area, an imbalance between feed supply and the nutritional demand of the area's domestic animals inevitably occurs during the annual winter and early spring (2, 3). Consequently, domestic herbivores experience weight loss, low milk production, and other issues (4). Being one of the most important alpine forages, *Elymus nutans* (*E. nutans*) is widely distributed in the QTP and is characterized by its high nutritional value and palatability for livestock. Therefore, the effective preservation of these local forages is becoming an increasingly prominent problem in alleviating the unbalanced feed supply in the cold alpine regions of QTP.

Ensiling is an efficient technique for the preservation of fresh feed and contributes to the continuous feeding of ruminants throughout the year in the QTP alpine regions (5). It has been well accepted that ensiling is an epiphytic lactic acid bacteria (LAB)-driven process and silage quality highly depends on their succession under anaerobic conditions (6, 7). In addition, previous studies have reported that the frequency and species of epiphytic LAB become more significant factors in determining the fermentation procedure (8, 9). During anaerobic fermentation, epiphytic LAB can ferment green forage water-soluble carbohydrates (WSCs) into organic acids for long-term preservation (10, 11). Thus, characterizing the plant surface microbial flora is essential for silage fermentation. However, the natural microbial population present in plants is influenced by various factors, such as crop species, climate, location, ultraviolet (UV) radiation intensity, and fertilizer application (8, 12). According to the information that we have, the QTP is characterized by unique geographical and ecological distinctions; accordingly, it harbors special microorganisms.

Recently, investigating the distribution patterns of attached microorganisms in plants in QTP has garnered growing attention due to their role in the forage ensiling process. Yang et al. (13) confirmed that altitude is a critical factor in shaping the epiphytic bacterial and fungal communities of *K. pygmaea*. Similarly, Ding et al. (14) also found an apparent difference in epiphytic microbial flora between *E. nutans* forages from the different grasslands of QTP, resulting in varying degrees of microbial community dynamics and fermentation quality for *E. nutans* silages when ensiled at an established temperature (~25°C). Nevertheless, there is still a lack of in-depth research on the distribution pattern of epiphytic microbial communities in forages and their subsequent fermentation quality of silage reserved at regional room temperature with the increased altitudes in QTP. Therefore, we presume that the altitudes (climatic conditions) of different regions in the QTP will affect the composition of microbial numbers and species attached to the forages, subsequently influencing the silage quality (13, 14). Thus, epiphytic microbial community composition structure surveys will provide information on the major and predominant microorganisms in forages along altitudinal gradients, which could help reveal their response to the fermentation quality of silage formulated from distinctive alpine regions of the QTP.

Hence, this study aimed to investigate the response of the epiphytic bacterial community, dynamics of bacterial changes and their functional shifts, and fermentation quality of *E. nutans* to altitudinal gradients in different alpine regions of QTP. The goal is to provide theoretical guidance for silage production in this region.

## RESULTS

### Microbial population and chemical characteristics of *E. nutans* before ensiling

As shown in Table 1, the numbers of LAB and yeast in fresh *E. nutans* from these three regions linearly decreased with increasing altitude, ranging from 7.14 to 5.34 and 5.41 to 3.9 $\log_{10}$ colony-forming units (cfu)/g fresh weight (FW), respectively. The lowest counts of LAB and yeast were found in the sample from Chenduo (CD) ($P < 0.001$). Mold was not detected in any of the fresh *E. nutans* samples. The dry matter (DM) contents in all fresh samples were approximately 400 g/kg, and no significant differences were

**TABLE 1** Microbial population and chemical composition of fresh *E. nutans* from different alpine regions

| Item[a] | Region[b,d] | | | SEM[c] | P-value | |
|---|---|---|---|---|---|---|
| | HY | XH | CD | | Linear | Quadratic |
| Microbial population | | | | | | |
| LAB, $\log_{10}$ cfu/g FW | 7.14[a] | 6.58[b] | 5.34[c] | 0.27 | <0.001 | 0.003 |
| Yeast, $\log_{10}$ cfu/g FW | 5.41[a] | 4.42[b] | 3.9[c] | 0.23 | <0.001 | 0.184 |
| Mold, cfu/g FW | ND | ND | ND | | | |
| Chemical composition | | | | | | |
| DM, g/kg | 404.3 | 400.6 | 402.4 | 3.51 | 0.849 | 0.762 |
| CP, g/kg DM | 74.8[b] | 75.6[b] | 85.1[a] | 1.82 | 0.003 | 0.059 |
| WSC, g/kg DM | 124.8 | 181.3 | 197.6 | 11.18 | <0.001 | 0.004 |
| aNDF, g/kg DM | 642 | 632 | 614 | 6.02 | 0.058 | 0.697 |
| ADF, g/kg DM | 331 | 331 | 308 | 5.19 | 0.051 | 0.215 |

[a]LAB, lactic acid bacteria; cfu, colony-forming unit; FW, fresh weight; DM, dry matter; CP, crude protein; WSC, water-soluble carbohydrate; aNDF, neutral detergent fiber; ADF, acid detergent fiber.
[b]HY, *E. nutans* collected from the grasslands of Huangyuan; XH, *E. nutans* collected from the grasslands of Xinghai; CD, *E. nutans* collected from the grasslands of Chenduo.
[c]SEM, standard error of the mean.
[d]Different lowercase letters (a–c) indicate significant differences between different silage groups (P < 0.05). ND, not detected.

observed (P > 0.05). The crude protein (CP) contents in fresh *E. nutans* forages from different locations varied from 74.8 to 85.1 g/kg DM, with the sample from CD county having a significantly higher value than the sample from Huangyuan (HY) and Xinghai (XH). Additionally, the WSC content linearly increased with altitude, varying from 124.8 to 197.6 g/kg DM (P < 0.001). The neutral detergent fiber (aNDF) and acid detergent fiber (ADF) contents exhibited a declining trend with the rising of the elevation, with no substantial differences among the different *E. nutans* forage groups (P > 0.05).

## Chemical and fermentation properties of *E. nutans* silages

The chemical composition of silage samples from three alpine regions after 60 days of fermentation is shown in Table 2. The DM content of *E. nutans* silage linearly decreased with increasing altitude in the order of HY (358.3 g/kg) > XH (348.4 g/kg) > CD (333.3 g/kg) (P < 0.01). Compared with fresh forages, the DM contents of *E. nutans* silages from HY, XH, and CD decreased by 11.4%, 13.0%, and 17.2%, respectively. However, WSC contents for all *E. nutans* silages exhibited an opposite trend to DM contents, with the lowest reduction percentage observed in *E. nutans* silage from CD (48.8%). Moreover, *E. nutans* silage from CD was characterized by a higher concentration of CP (92.0 g/kg DM), and there was a linear decrease in the ammonia nitrogen ($NH_3$-N) concentrations with increasing elevation (P < 0.01). The aNDF contents in silage samples from these three regions ranged from 593.4 to 612.6 g/kg DM, with no significant differences, while the

**TABLE 2** Chemical characteristics of *E. nutans* silages prepared from different alpine regions after 60 days of ensiling

| Item[a] | Region[b,d] | | | SEM[c] | P-value | |
|---|---|---|---|---|---|---|
| | HY | XH | CD | | Linear | Quadratic |
| DM, g/kg | 358.3[a] | 348.4[a] | 333.3[b] | 4.18 | 0.005 | 0.623 |
| WSC, g/kg DM | 33.2[c] | 68.8[b] | 101.2[a] | 9.39 | <0.001 | 0.595 |
| CP, g/kg DM | 84.1[b] | 83.4[b] | 92.0[a] | 1.63 | 0.017 | 0.067 |
| $NH_3$-N, g/kg TN | 75.12[a] | 70.29[ab] | 67.55[b] | 1.29 | 0.007 | 0.552 |
| aNDF, g/kg DM | 608.5 | 612.6 | 593.4 | 4.21 | 0.128 | 0.168 |
| ADF, g/kg DM | 336.5[a] | 321.3[b] | 297.7[c] | 5.91 | <0.001 | 0.352 |

[a]DM, dry matter; WSC, water-soluble carbohydrate; CP, crude protein; $NH_3$-N, ammonia nitrogen; TN, total nitrogen; aNDF, neutral detergent fiber; ADF, acid detergent fiber.
[b]HY, *E. nutans* collected from the grasslands of Huangyuan; XH, *E. nutans* collected from the grasslands of Xinghai; CD, *E. nutans* collected from the grasslands of Chenduo.
[c]SEM, standard error of the mean.
[d]Different lowercase letters (a–c) indicate significant differences between different silage groups (P < 0.05).

ADF contents of silage samples decreased linearly with increasing altitude ($P < 0.001$), and the highest decrease (3.3%) appeared in *E. nutans* silage from CD.

The dynamics of fermentation parameters are shown in Table 3. The interactions between region and silage ensiling days showed significant effects ($P < 0.001$) on pH, lactic acid (LA), acetic acid (AA), and the ratio of lactic and acetic acid (LA/AA). At 7 days of ensiling, there were no differences among the pH values of the three silage groups. Subsequently, the pH of *E. nutans* silage from CD rapidly decreased from 6.34 to 5.82 within the initial 14 days of fermentation, remaining the lowest thereafter. Although the pH of the silage sample from HY dropped below 6 by the end of ensiling, it was still significantly higher than XH (4.54) and CD (4.44). During the whole ensiling process, the lactic acid concentrations in *E. nutans* silage samples from CD and XH were higher than in the sample from HY. The concentrations of AA for all silages increased as the ensiling day progressed. However, the lowest concentration of AA was observed in *E. nutans* silage sample from XH (5.73 g/kg DM) after 60 days of ensiling. The value of LA/AA remained above 1 throughout the entire ensiling period, and LA/AA was consistently higher in silage samples from XH (1.91–4.38) and CD (1.90–4.09) as compared to the silage sample from HY (1.22–2.01).

## Dynamics of bacterial community composition in *E. nutans* silages

As indicated by Shannon indices, alpha diversity is displayed in Fig. 1A. The results revealed that alpha diversity both decreased in fresh and ensiled *E. nutans* forages (after 60 days of fermentation) from XH and CD than in samples from HY. Based on the result of the principal coordinate analysis (PCoA), distinct differences in the progression of bacterial communities among silage samples from different regions were observed. The bacterial communities of *E. nutans* silage samples from HY were not separated throughout the whole fermentation process. However, the bacterial communities in

**TABLE 3** Dynamics of fermentation characteristics in *E. nutans* silage prepared from different alpine regions

| Item[a] | Ensiling days | Region[b,e] | | | SEM[c] | P value[d] | | |
|---|---|---|---|---|---|---|---|---|
| | | HY | XH | CD | | R | D | R × D |
| | 7 | 6.26[A] | 6.24[A] | 6.34[A] | 0.030 | <0.001 | <0.001 | <0.001 |
| | 14 | 6.24[Aa] | 6.04[Aab] | 5.82[Bb] | | | | |
| | 30 | 5.86[Ba] | 4.93[Bb] | 4.59[Cc] | | | | |
| | 60 | 5.55[Ba] | 4.54[Bb] | 4.44[Cb] | | | | |
| pH | avg | 5.98 | 5.44 | 5.30 | | | | |
| | 7 | 4.38[Db] | 6.64[Da] | 6.14[Dab] | 0.189 | <0.001 | <0.001 | <0.001 |
| | 14 | 7.19[Cb] | 18.4[Ca] | 16.5[Ca] | | | | |
| | 30 | 13.0[Bc] | 26.5[Bb] | 34.3[Ba] | | | | |
| | 60 | 15.7[Ac] | 31.7[Ab] | 46.7[Aa] | | | | |
| Lactic acid, g/kg DM | avg | 10.1 | 20.8 | 25.91 | | | | |
| | 7 | 3.59[C] | 3.48[D] | 3.27[C] | 0.065 | <0.001 | <0.001 | <0.001 |
| | 14 | 3.05[Cc] | 4.02[Cb] | 4.74[Ba] | | | | |
| | 30 | 5.49[Bab] | 4.85[Bb] | 5.85[Ba] | | | | |
| | 60 | 7.51[Ab] | 5.73[Ac] | 9.23[Aa] | | | | |
| Acetic acid, g/kg DM | avg | 4.91 | 4.52 | 5.77 | | | | |
| | 7 | 1.22[Bb] | 1.91[Ca] | 1.90[Ca] | 0.049 | <0.001 | <0.001 | <0.001 |
| | 14 | 2.36[Ac] | 4.59[Ba] | 3.49[Bb] | | | | |
| | 30 | 2.36[Ab] | 5.46[Aa] | 5.89[Aa] | | | | |
| LA/AA | 60 | 2.09[Ab] | 5.54[Aa] | 5.07[Aa] | | | | |
| | avg | 2.01 | 4.38 | 4.09 | | | | |

[a]DM, dry matter; LA/AA, the ratio of lactic acid and acetic acid.
[b]HY, *E. nutans* collected from the grasslands of Huangyuan; XH, *E. nutans* collected from the grasslands of Xinghai; CD, *E. nutans* collected from the grasslands of Chenduo.
[c]SEM, standard error of the mean.
[d]R, region; D, ensiling day; R × D, interaction between region and ensiling day.
[e]Different lowercase letters (a–c) indicate significant differences between different silage groups ($P < 0.05$). Different uppercase letters (A–D) indicate significant differences between ensiling days ($P < 0.05$).

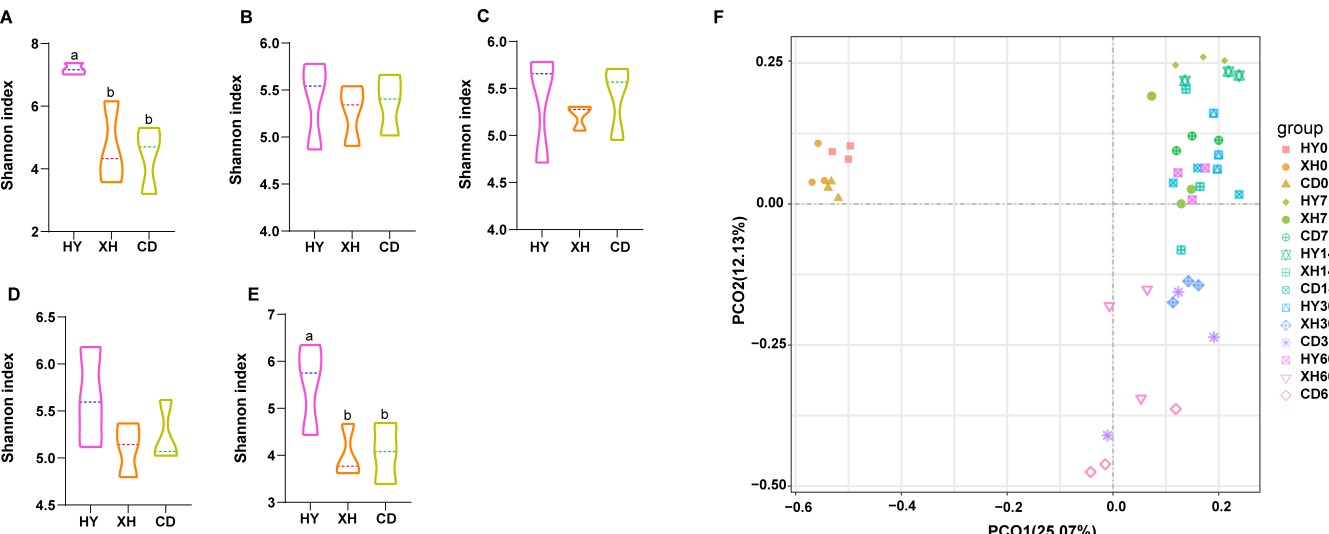

**FIG 1** Bacterial community diversities of fresh and ensiled *E. nutans* collected from different regions. Arabic number indicates days of ensiling. Different lowercase letters (a and b) indicate significant differences between different silage groups (*P* < 0.05). (A–E) Alpha diversities (Shannon indexes) of bacterial community in fresh and ensiled *E. nutans* after 7, 14, 30, and 60 days of fermentation. (F) Principal coordinate analysis (PCoA) of bacterial community in *E. nutans* silages after 0, 7, 14, 30, and 60 days of fermentation. HY, *E. nutans* collected from the grasslands of Huangyuan County; XH, *E. nutans* collected from the grasslands of Xinghai County; CD, *E. nutans* collected from the grasslands of Chenduo County.

silage samples from XH and CD were clustered from 7 to 14 days and 30 to 60 days, respectively.

The dynamics of bacterial community structure and succession in *E. nutans* silages from different regions are shown in Fig. 2 at the genus and species levels. The major epiphytic bacteria at the genus level (Fig. 2A) for fresh samples from these three regions were *Pseudomonas*, *Sphingomonas*, and *Xanthomonas*. A fresh sample from HY was found to be dominated by *Pseudomonas* and *Sphingomonas*, whereas in XH, *Pseudomonas* and *Xanthomonas* were the most abundant, with 30.09% and 21.64% of the total population, respectively. In contrast, *Pseudomonas* was the most prevalent organism in the CD sample, reaching a percentage of 39.88%. Notably, desirable *Lactobacillus* accounted for less than 1% in all fresh *E. nutans* forages. The undesirable genus including *Pantoea* (*Pantoea agglomerans*), *Hafnia* (*Hafnia* sp.), and *Rahnella* (*Rahnella* sp.) were predominant in all silage samples ensiled for 7 days. However, with the extension of ensiling time, their abundance gradually decreased, and even after 60 days of fermentation, they made up less than 14% of the population. As expected, nine species of LAB including *Lactiplantibacillus plantarum*, *Pediococcus acidilactici*, *Weissella* sp., and *Leuconostoc mesenteroides* were identified in this study. Among these species, *Lactiplantibacillus plantarum* and *Pediococcus acidilactici* became the prevalent species in silage samples from XH and CD from 30 to 60 days of ensiling. Particularly, the proportion of *Lactiplantibacillus plantarum* in *E. nutans* silage from CD exceeded 30% after 60 days of fermentation (Fig. 2B). Moreover, we also found that the abundance of *Weissella* sp. in the silage sample from CD reached at 13.23% when ensiled for 7 days; however, such a high abundance was not found in other two groups of silage throughout the ensiling process. During the ensiling process, we also found a certain amount of *Serratia* sp. and *Leuconostoc mesenteroides* present in all groups. Meanwhile, *Staphylococcus epidermidis* was prevalent in silage samples from XH and CD; *Latilactobacillus sakei*, in silage sample from CD; *Enterococcus mundtii*, in silage sample from HY; and *Clostridium perfringens* and *Clostridium tyrobutyricum*, in silage sample from XH. From the stream graph (Fig. 2C through E), it is clear that different regions with varying altitudes had a significant impact on the succession of bacterial communities in *E. nutans* silages.

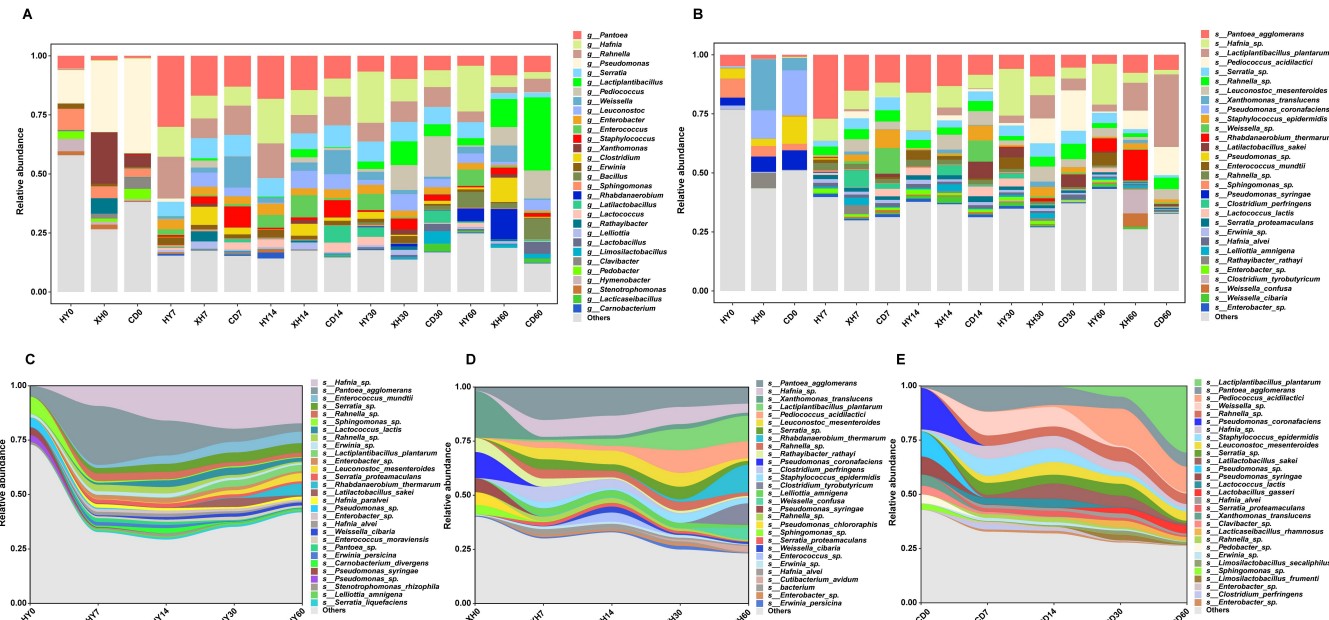

**FIG 2** Bacterial community compositions and successions *E. nutans* silages prepared from different regions. Arabic number indicates days of ensiling. (A and B) Relative abundance of bacterial community at the genus and species level, respectively, across different groups and fermentation times. (C–E) Bacterial community successions are aggregated and colored by genus on a stream graph for *E. nutans* silages prepared from HY, XH, and CD, respectively. HY, *E. nutans* collected from the grasslands of Huangyuan County; XH, *E. nutans* collected from the grasslands of Xinghai County; CD, *E. nutans* collected from the grasslands of Chenduo County.

To explore the differentially abundant taxa among the silage groups from different regions, a more rigorous linear discriminant analysis effect size analysis was performed (Fig. 3). The results showed that *Clavibacter* and *Agreia* were differentially abundant taxa in *E. nutans* forage from CD, while *Zoogloea*, *Rahnella*, and *Curtobacterium* were observed in forage from HY, and *Xanthomonas* was in forage from XH. After 7 days of ensiling, significantly different bacteria appeared in silage samples from CD and XH, including several genera or species of LAB. For instance, *Lacticaseibacillus* and *Weissella* sp. were differentially identified in the silage sample from CD, while *Lactiplantibacillus* was the differentially identified taxa in the silage sample from XH. However, *Pseudomonas* and *Clostridium perfringens* unexpectedly exhibited differentially abundant taxa in silage samples from HY and XH, respectively. From days 14 to 60 of ensiling, *Lacticaseibacillus rhamnosus* and *Levilactobacillus brevis* still appeared as the differentially abundant taxa in silage samples from CD.

## Bacterial cooccurrence, cooccurrence network complexity, and stability of *E. nutans* silages

The network of the bacterial community in silages from different regions showed different patterns of occurrence (Fig. 4). Topological parameters of the network, including the numbers of nodes and edges, as well as the degree of betweenness and assortativity, were used to evaluate the complexity of bacterial community network. The greater complexity of the network is indicated by higher numbers of nodes and edges and lower levels of betweenness and assortativity. Meanwhile, a higher ratio of negative to positive values represented high stability in the bacterial network. The results showed that the more stable bacterial community network evidenced by higher negative/positive ratio was found in the silage sample from CD versus that from XH and HY. Meanwhile, the dramatically reduced complexity of bacterial network was observed in silage sample from XH and CD, with fewer edge numbers and greater degrees of betweenness when compared with the silage sample from HY. Furthermore,

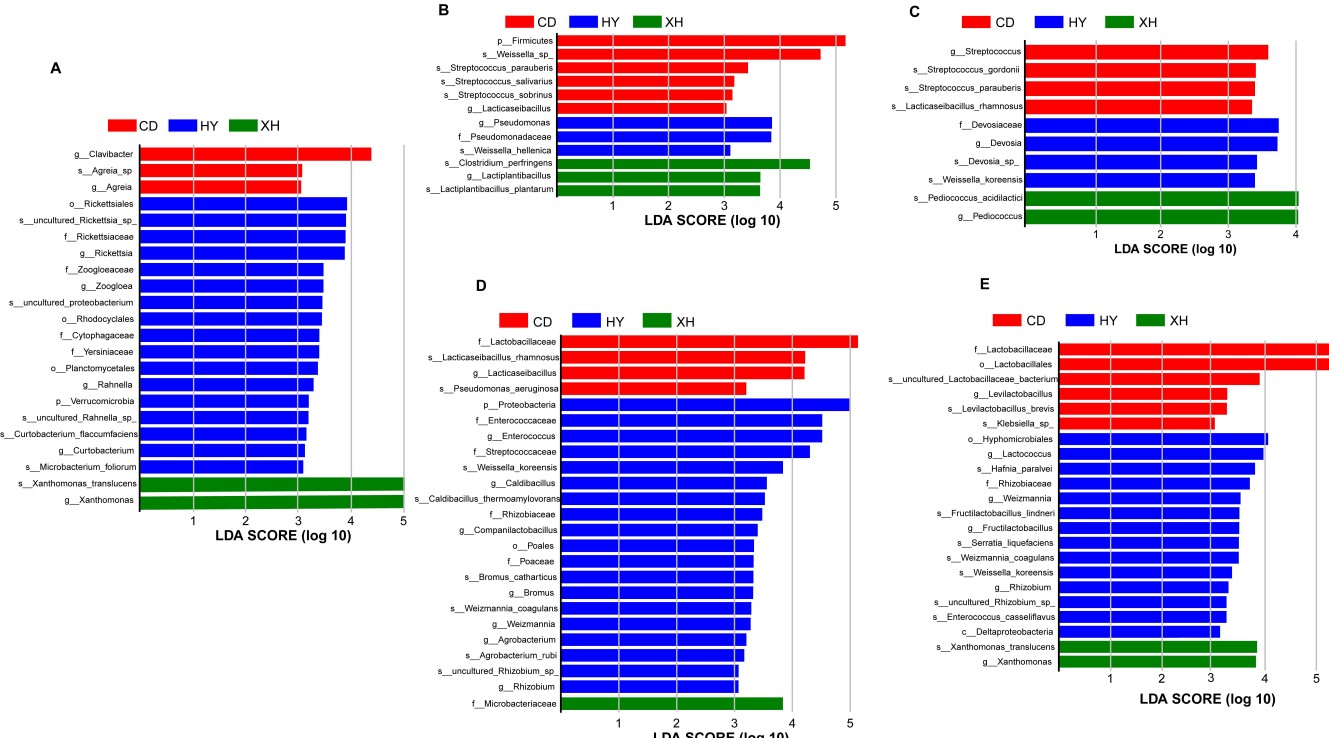

**FIG 3** The identified differential bacteria (*P* < 0.05, and linear discriminant analysis [LDA] score >3) among the three silage groups. HY, *E. nutans* forage samples from Huangyuan County; XH, *E. nutans* forage samples from Xinghai County; CD, *E. nutans* forage samples from Chenduo County. (A) Differential bacteria identified in fresh *E. nutans* forages. (B) Differential bacteria identified in silages ensiled for 7 days. (C) Differential bacteria identified in silages ensiled for 14 days. (D) Differential bacteria identified in silages ensiled for 30 days. (E) Differential bacteria identified in silages ensiled for 60 days.

we noticed that the higher relative abundance of *Lactiplantibacillus* in CD resulted in negative correlations (green line represents negative correlation) only with *Paenibacillus*. In contrast, the negative correlations between *Lactiplantibacillus* and other bacterial genus were more complex in the silage samples from XH and HY, in particular for HY. This may be helpful in explaining the simpler complexity of bacterial community network in silage sample from CD, which is led by a higher abundance of *Lactiplantibacillus*, weakening the competitiveness of epiphytic bacteria during the ensiling.

## Correlation analysis between abiotic factors and microorganisms of *E. nutans*

The redundancy analysis (RDA) was performed to investigate the potential abiotic factors differentiating microbial community structures of fresh and ensiled *E. nutans*. As shown in Fig. 5A, eight environmental factors including altitude, WSC, CP, DM, NDF, ADF, and numbers of yeast and LAB together explained 59.00% and 35.11% of the total variance by RDA axes 1 and 2, respectively. Among these environmental factors, WSC and yeast number had a significant effect on the distribution of epiphytic bacteria. The predominant genera *Pseudomonas* and *Xanthomonas* showed positive correlations with WSC, while they had negative correlations with yeast. In comparison, bacterial community compositions in *E. nutans* silage were significantly associated with altitude, WSC, LA, and ammonia nitrogen ($R^2$ = 0.8964, $P$ = 0.001; $R^2$ = 0.8527, $P$ = 0.003; $R^2$ = 0.8920, $P$ = 0.001; $R^2$ = 0.7180, $P$ = 0.021). These environmental variables could explain up to 62% (35.97% and 26.42% on RDA axes 1 and 2, respectively) of the total variability (Fig. 5B). *Lactiplantibacillus* and *Pediococcus* showed positive correlations with LA, WSC, and altitude. Meanwhile, *Hafnia*, *Pantoea*, and *Serratia* were also positively associated with NH$_3$-N and pH.

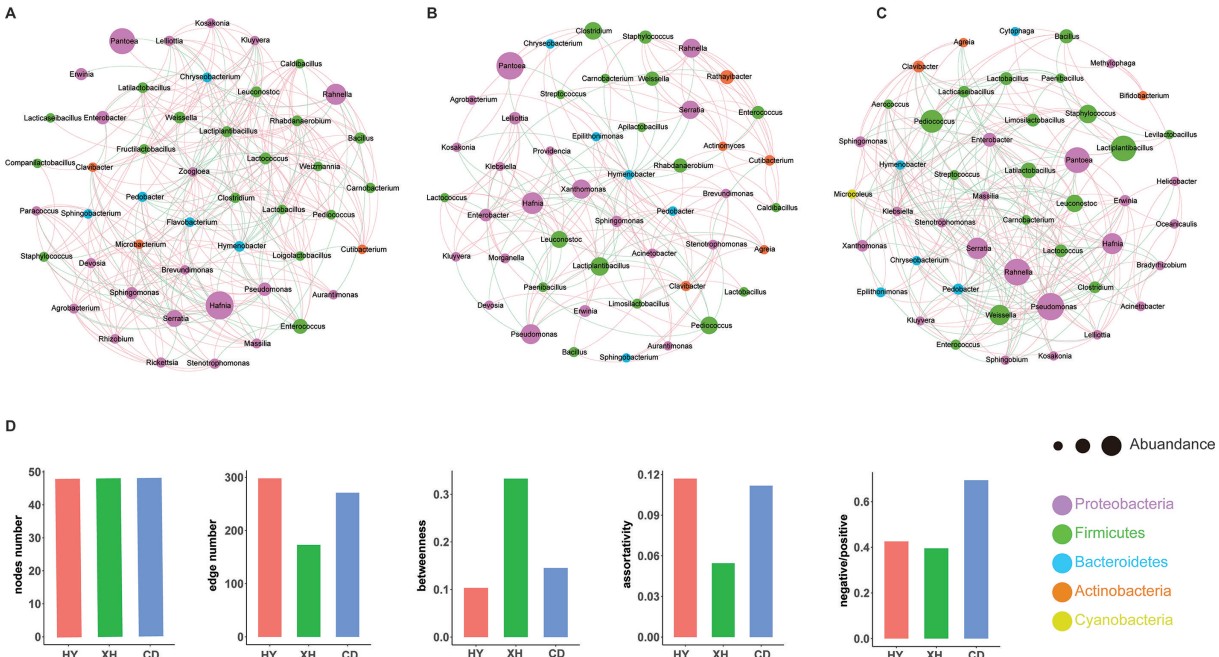

**FIG 4** Bacterial cooccurrence network analyses of *E. nutans* silage prepared from different alpine regions. (A–C) Bacterial cooccurrence networks of *E. nutans* silage prepared from Huangyuan (HY), Xinghai (XH), and Chenduo (CD) counties. (D) Numbers of nodes and edges, degrees of betweenness and assortativity, and negative/positive correlation ratios of silage bacterial cooccurrence patterns. Node colors indicate bacterial genera, and node sizes indicate their relative abundance. Edges are colored according to negative (green) and positive (red) correlations.

## Dynamics of predicted metabolic functions of the bacterial community in the *E. nutans* silages

Silage fermentation process is facilitated by microbial activities via complex metabolic pathways, with degrading substrates or converting metabolites. Predicting the microbial communities' functions allows us to evaluate the influence of bacterial communities on the variations in the metabolic pathways underlying silage fermentation process. Therefore, we used the Kyoto Encyclopedia of Genes and Genomes (KEGG) database with PICRUSt to predict the metabolic pathways of silages from different regions. As shown in Fig. 6, the predicted functions of bacterial communities were categorized into cellular processes, environmental information processing, genetic information processing, human diseases, metabolism, and organism system for *E. nutans* silages from different regions during the fermentation process (Fig. 6A). The heatmap of functional profiles reveals the functional shifts of bacterial communities in *E. nutans* silages with the progressed fermentation time (Fig. 6B through F). Among these predicted functions, we observed remarkable upregulations of carbohydrate metabolism, nucleotide metabolism, replication and repair, and translation in the silage sample from CD compared with the other two silage groups throughout all ensiling periods. Specifically, at 7 or 14 days of ensiling, a higher abundance of amino acid metabolism was observed in silages from CD than the other two silage groups. However, at the final stage of fermentation, amino acid metabolism and metabolism of cofactors and vitamins were markedly down-regulated in silage samples from CD. Additionally, energy metabolism in *E. nutans* silage from CD consistently remained at a lower abundance compared to the other three silage groups throughout the entire fermentation process.

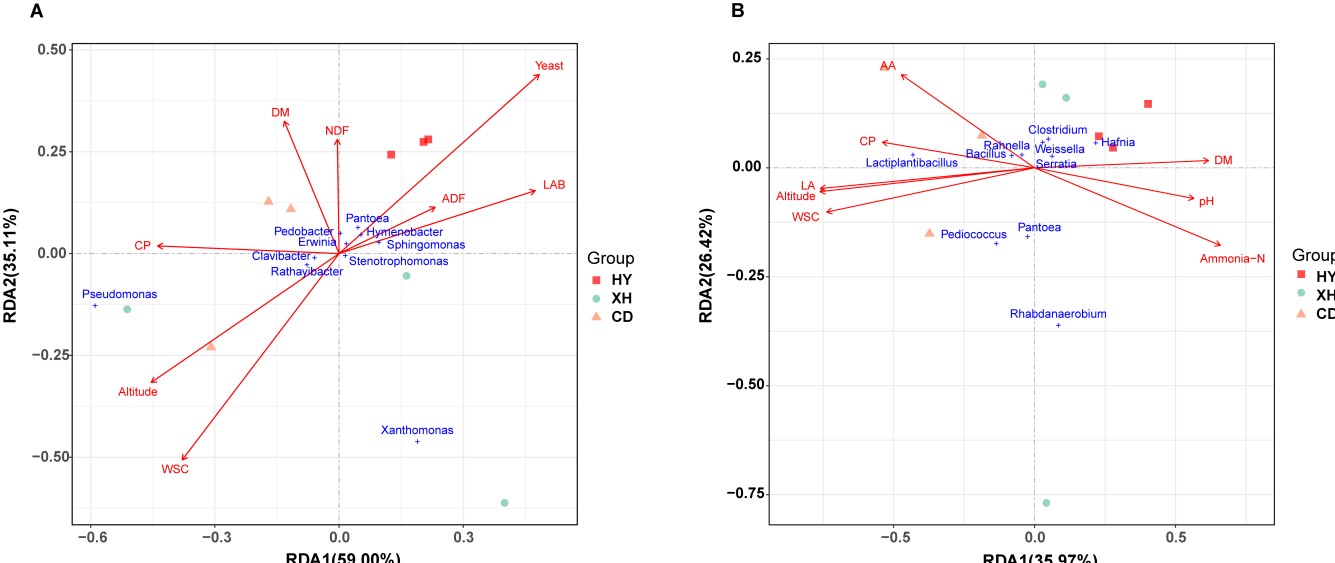

**FIG 5** Redundancy analysis of environmental factors and bacterial community of *E. nutans* forage along the elevation gradient. (A) Correlation between the environmental factors and bacterial community in fresh *E. nutans*. (B) Correlation between the environmental factors and bacterial community in *E. nutans* silages after 60 days of fermentation. The axes are labeled with the percentage of total variance explained (%). HY, forage sample collected from Huangyuan County; XH, forage sample collected from Xinghai County; CD, forage sample collected from Chenduo County. DM, dry matter; WSC, water-soluble carbohydrate; CP, crude protein; NH$_3$-N, ammonia nitrogen; aNDF, neutral detergent fiber; ADF, acid detergent fiber; LAB, lactic acid bacteria; LA, lactic acid; AA, acetic acid.

## DISCUSSION

### Characteristics of fresh *E. nutans* from different regions

Silage production depends on controllable and uncontrollable factors. The controllable factors that farmers control are growth stage of the silage material, moisture content at harvest, silage methods, and use of silage additives. Although uncontrollable climate-related factors, such as temperature, average rainfall, and soil fertility, characterize hot or cold regions, they can have negative effects. The influence of soil fertility on the chemical composition of crop has been recognized for many years (15). Additionally, forages in cold regions have a short growing season and low temperatures, resulting in lower nutritive value compared to the regions with more favorable growing conditions (16). As we know, the QTP is known for its harsh environmental conditions, including low temperatures, strong UV radiation, and hypoxia (13), which can greatly affect the epiphytic microbiota and chemical properties of plants that survive in this region. This study observed an increase in the content of WSC and CP in fresh *E. nutans* as the altitude increased, which may be due to the decreased respiration of plants at higher elevations allowing for the accumulation of WSC, CP, and crude fat in the cell protoplasm of plants (13). Similarly, Ding et al. (14) also found that CP and WSC in *E. nutans* increased along with altitudinal gradients. The epiphytic LAB numbers of all *E. nutans* forages were higher than 5 log$_{10}$ cfu/g FW; however, their numbers decreased with the increase in altitude, reflecting the challenging conditions at higher elevations. In similar, Yang et al. (13) also reported that all fresh *K. pygmaea* from the highest altitude (F5000) had the lowest abundances of LAB, *Enterobacteriaceae*, and aerobic bacteria. Therefore, the present result suggested that the increasing altitude is accompanied by increased physiological challenges, resulting in an unfavorable environment for the growth of these epiphytic microflora distributed on the surface of *E. nutans*.

### Silage characteristics of *E. nutans* from different regions

Silage fermentation is an epiphytic microorganism-driven process, which leads to organic acid accumulation and a decrease in pH. This study showed rapid LA

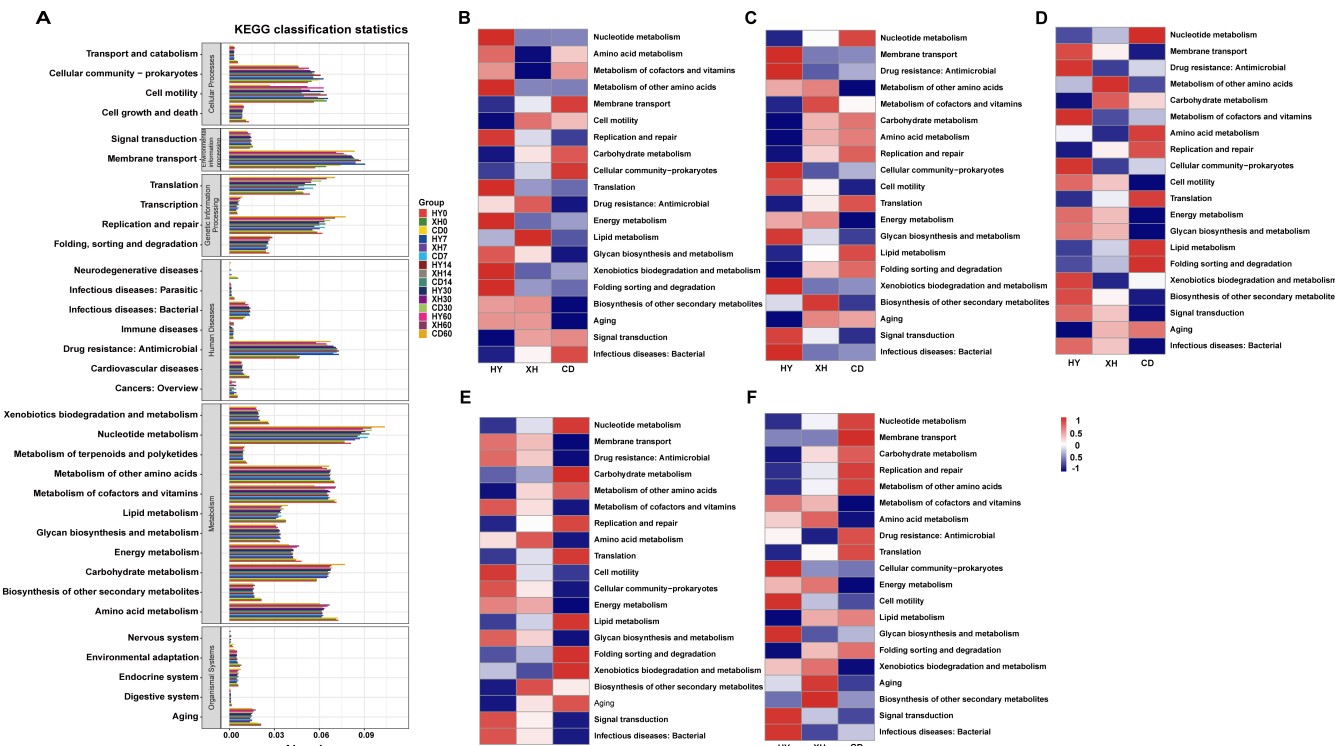

**FIG 6** Function profiles of bacterial community in different groups. (A) Summary of second level of Kyoto Encyclopedia of Genes and Genomes (KEGG) orthologue functional predictions (top 20 abundant functions) explained by PICRUSt2. Arabic number indicates days of ensiling. Heatmap of 16S rRNA gene-predicted KEGG function profiles for *E. nutans* forage ensiled for 0 day (B), ensiled for 7 days (C), ensiled for 14 days (D), ensiled for 30 days (E), and ensiled for 60 days (F).

accumulation and pH decline in *E. nutans* silage from CD after 7 days of ensiling, primarily due to high WSC content, which provided substrates for LAB proliferation (17). Meanwhile, LAB naturally present in forage from high-altitude environments possess a strong capability to adapt to the fluctuating conditions of this harsh environment, and this may be conducive to forage ensiling (18). The previous study stated that the ratio of LA and AA (LA/AA) is indicative of adequate silage fermentation, and well-preserved silage is typically characterized by a value of 2.5 to 3.0 for LA/AA (19). The LA/AA of *E. nutans* silage from HY ranged from 1.22 to 2.09 throughout the fermentation process, confirming poor fermentation quality. In short, the dynamic variations of these fermentation parameters indicate that the fermentation of *E. nutans* silage reached a relatively stable status in the later stage of ensiling (from the ensiling period of 30 to 60 days) and the fermentation qualities were very different among these silage groups.

Ammonia nitrogen, as a decisive indicator of protein degradation, is often used to evaluate silage quality. Ding et al. (14) observed decreased $NH_3$-N concentration and improved CP content in *E. nutans* silage along the elevational gradients. In agreement, we found that the contents of CP increased in *E. nutans* silage, while the amount of $NH_3$-N decreased with the altitudinal gradients. Expectedly, among these silage groups, the lowest amount of $NH_3$-N associated with the lowest pH presented in *E. nutans* silage from CD, which was detrimental to the growth and proliferation of undesirable microorganisms with proteolytic activity like *Clostridium* and *Enterobacter* (20).

## Microbial community structure of *E. nutans* silage from different regions

Natural silage fermentation heavily depends on the epiphytic microflora; especially, epiphytic LAB is responsible for pH decline and the transition to an anaerobic and acidic environment. The lower alpha diversities of *E. nutans* forages from CD and XH,

compared to those in the raw material from HY, are similar to the changes of LAB and yeast counts observed with increasing elevation. The PCoA plot was created to illustrate the differences in bacterial community structure between different silages from different regions. The distinct clustering between the fresh and ensiled *E. nutans* was attributed to the disappearance of aerobic and acid-intolerant microorganisms during the silage fermentation process (21). Simultaneously, a clear separation between the symbols of HY and XH (or CD) persisted throughout the ensiling period. This could be explained by the variations in chemical compositions and epiphytic microbial flora of *E. nutans* forages collected from different regions, influencing shifts in the microbial community. The reduction in the distance between the symbols of XH and CD from days 7 to 60 suggests that the bacterial community composition tended to become more similar as ensiling progressed, thereby explaining the similar fermentation quality of *E. nutans* silages in XH and CD.

Different bacterial community structures were observed in pre- and post-silages. Many previous studies concluded that bacterial colonization of plant surfaces depends on many factors, such as plant species, environmental conditions, time period, geographical location, the intensity of solar radiation, and the type of fertilizer applied (8, 22). Based on the results of the correlation analysis between environmental factors and bacterial community, we confirmed that the microbial population and nutritional value of *E. nutans* were significantly affected by altitude distribution. Substantially, enhanced fermentation quality and higher abundance of wanted microbes (like *Lactiplantibacillus* and *Pediococcus*) were observed along the elevational gradient.

In this study, the major bacterial genera identified were *Pseudomonas*, *Xanthomonas*, and *Sphingomonas* for fresh *E. nutans*, which slightly differed from the major epiphytic bacteria, such as *Enterobacter*, *Leuconostoc*, *Weissella*, *Lactobacillus*, and *Pseudomonas*, reported in fresh *E. nutans* collected from different regions of QTP (14). Among these, *Xanthomonas* is a well-studied plant-associated Gram-negative bacteria responsible for many crop diseases and economically crop yield losses (23). Similarly, *Pseudomonas* is also a well-known pathogen causing a diversity of healthcare-associated infections, which could be found on a wide variety of environment such as inanimate surfaces, human body, and high-altitude soil and lakes (24, 25). Therefore, the biocontrol of these pathogen has raised much attentions in agriculture area. The abundance of *Pseudomonas* in this study increased with the elevation increased, confirming once again its cold temperature-tolerant ability to adapt to the frigid environment of CD with high altitude. After 60 days of ensiling, alpha diversity in the silage sample from HY was also higher than that in samples from XH and CD, which was consistent with the highest pH of *E. nutans* silage from HY than the other two regions. Compared with the initial stage of ensiling, the clear shift of the bacterial community from *Pseudomonas coronafaciens*, *Sphingomonas* sp., and *Xanthomonas translucens* to *Lactiplantibacillus plantarum* and *Pediococcus acidilactici* occurred in ensiled *E. nutans* from XH and CD at the final stage of fermentation. *Lactiplantibacillus* and *Pediococcus* are generally desirable during the ensiling process because they can convert plant carbohydrates to lactic acid, thereby lowering the pH value of the silage. Therefore, in this study, the higher relative abundances of *Lactiplantibacillus* and *Pediococcus* in *E. nutans* silages from XH and CD were the main reason for their superior fermentation quality over silage from HY. In contrast, *Pantoea agglomerans*, *Hafnia* sp., and *Serratia* sp. were still the predominant species in silage from HY after 60 days of ensiling. As is well known, *Hafnia* and *Serratia* as members of *Enterobacteriaceae* are acid intolerant (26), and they have proteolytic activities to deaminate some amino acids to ammonia nitrogen (27). This may be the reason why the higher amount of $NH_3$-N appeared in silage sample from HY rather than other two regions.

## Co-occurrences of bacterial community for *E. nutans* silages

The epiphytic bacterial community functioned as an initiator in the silage ecosystem, and the interactions among these microorganisms were very complex. The co-occurrence

networks constructed in this study clearly demonstrated the interactions between epiphytic and silage microbiomes from different regions with varying altitudes. A greater node or edge number indicated a higher network complexity. Consistent with the findings of Zhao et al. (21), our current study identified a lower edge number which was associated with better fermentation quality in silage samples from XH or CD. This might be attributed to the suppression of unwanted microorganisms and the lower alpha diversity in silages from XH or CD compared to those in silage from HY. In contrast, a higher negative/positive value represents a more stable bacterial network structure and reduced competition among the bacterial communities. In the current study, the silage sample from CD had a more stable microbial network structure than the silage samples from the other two regions. Banerjee et al. (28) ever concluded that the competition among the microbial communities was alleviated with the addition of straw and nutrients to the soil. Therefore, we speculated that the enhanced stability of the bacterial network in *E. nutans* from CD could be attributed to the nutritional factors, especially WSC or CP which might reduce the competition among the microbiome to improve the stability of the network. In summary, the results of microbial co-occurrence suggested that bacterial community structure of *E. nutans* silages was markedly altered in response to the increasing altitudes.

## Functional shifts of bacterial community for *E. nutans* silages

Silage fermentation quality and flavor were primarily facilitated by the metabolisms of amino acids and carbohydrates due to protein hydrolysis and sugar degradation throughout the ensiling process (29). In this study, carbohydrate metabolism in silage sample from CD showed higher activity throughout almost entire fermentation process compared with that in silage samples from HY and XH. This is evidenced by the intensive lactic acid fermentation by the vigorous proliferation of LAB in silage from CD. As an essential metabolic pathway, the promoted amino acid metabolism is mainly related to the proteolysis induced by undesirable microbes (30). This is to be expected; at the final stage of ensiling (30 to 60 days of ensiling), amino acid metabolism showed a decreasing trend with the increasing elevations, which was consistent with the increase in CP contents. Nucleotides provide major energy for cellular processes and are commonly used to synthesize DNA (21). Interestingly, nucleotide metabolism remained at a higher abundance in silage from CD than that in silage samples from the other two regions during the whole fermentation process. This was in contrast to the trends observed in energy metabolism, consistent with the shifts in genetic information processing including translation and replication and repair. Based on the descriptions of previous finding, these upregulated genetic functions are likely caused by the long-term acid stress in silage (7). Therefore, the significant upregulation of nucleotide metabolism may be an attempt to provide energy for genetic information processing.

## Conclusions

With the increasing elevations, the diversity of epiphytic microbiota and the counts of LAB and yeast in *E. nutans* decreased. However, the nutritional components of forages significantly improved. After 60 days of ensiling, a more stable bacterial network structure was observed, along with a higher abundance of fermentation-promoting LAB including *Lactiplantibacillus* and *Pediococcus* in *E. nutans* silages from CD. Additionally, the fermentation quality was improved, as evidenced by higher lactic acid concentrations and lower levels of ammonia nitrogen, amino acid metabolism, and pH in ensiled *E. nutans* from CD. In summary, altitude is a significant factor that influences the epiphytic microbial structure and nutrition distribution of *E. nutans* forage, thus leading to varying degrees of fermentation. Therefore, it is highly recommended to inoculate LAB that aid fermentation, which have been screened from high-altitude areas, for *E. nutans* silage making in the Qinghai region, especially for the regions with low altitudes.

## MATERIALS AND METHODS

### Silage preparation

The *E. nutans* grasses approximately at a similar growth stage were harvested from three different grasslands of Huangyuan, Xinghai, and Chenduo counties of Qinghai province on 13 to 21 August 2021. The *E. nutans* grasses were randomly collected from three sampling sites in each grassland. The altitudes of the three sampling sites were 2,567, 2,610, and 2,662 m, respectively, for Huangyuan County. The altitudes of the three sampling sites were 3,555, 3,634, and 3,647 m, respectively, for Xinghai County, while 4,550, 4,582, and 4,636 m, respectively, for Chenduo County. The collected materials were immediately transported to the local laboratory and cut into 2-cm lengths using a paper cutter. After through mixing, approximately $500 \pm 10$ g of chopped material was packed into a polyethylene plastic bag (silo, 35 cm × 50 cm) and vacuum sealed. Overall, 144 bags (3 plots × 3 sampling sites × 4 replicates × 4 ensiling times) were prepared and stored at local room temperature for 60 days. Silage samples were collected to analyze fermentation characteristics and microbial community after 7, 14, 30, and 60 days of ensiling, while the chemical composition of *E. nutans* silage was determined after 60 days of fermentation.

### Microbial population, fermentation profile, and chemical composition analyses

Twenty grams of fresh or green fodder were mixed with 180 mL of sterile water and homogenized in a juicer for 30 s, followed by filtering them through four layers of cheesecloth. An aliquot of the solution was consecutively diluted seven times for microbial counting using plate cultures. After incubation at 37°C for 72 h, the LAB were numbered on de Man, Rogosa, Sharpe agar. Yeasts and molds were counted on potato dextrose agar after incubation at 28°C for 48 h. An additional part of the filtrate was used for determining the concentrations of organic acids, $NH_3$-N, WSC, and pH. The concentrations of organic acids such as LA, AA, propionic acid, and butyric acid in the filtrate were determined as previously established (31). Concentrations of $NH_3$-N and WSC were detected using the method referred to by Broderick and Kang (32) and Thomas (33), respectively. The pH was determined using a pH meter with a glass electrode (Hanna Instruments Italia Srl, Padua, Italy).

To detect DM content, forage samples were placed in an oven and dried at 65°C for 72 h. All dried samples were passed through a 1-mm sieve to assess chemical composition. CP content of all forage samples was assessed by the Kjeldahl method (34), whereas aNDF and ADF contents were estimated using an Ankom 2000 fiber analyzer (35).

### Bacterial composition SMRT analyses

Total genomic DNA of all fresh and ensiled forages was isolated using a DNA isolation kit (DP302-02, Tiangen, China) according to the manufacturer's instructions. PCR amplification of the full-length 16S rRNA gene for single molecule real-time sequencing (SMRT) analyses was performed according to the description in the report of Li et al. (36). The PCR amplicons were purified and quantified using the PicoGreen dsDNA Assay kit (Invitrogen, Carlsbad, CA, USA). Sequencing of these amplicons was performed on a PacBio platform of Wuhan Frasergen Bioinformatics Co., Ltd. (Wuhan, China), and the generated circular consensus reads were processed as stated by Li et al. (36). Then, the representative sequences were annotated using the SILVA bacterial 16S rRNA database by a QIIME-based wrapper of RDP-classifier v.2.2 with a confidence cutoff of 0.8. The unique sequence was clustered into operational taxonomic units using UPARSE 7.1 with 97% sequence identity. Bacterial community composition was visualized at the genus and species levels using the SILVA v138 database. Shannon indices were calculated to assess alpha diversity. To examine the beta diversity between different treatments, we conducted a PCoA. We used PICRUSt2 to predict bacterial community functions based on the KEGG database. For visualizing bacterial community successions, we employed

a stream graph, which we generated based on the method described in the previous report (37). We constructed a co-occurrence network following the analytical methods outlined by Zhao et al. (21). To investigate the association between environmental factors and bacterial communities in forages, we performed an RDA.

## Statistical analysis

Before analysis, all microbial count data were log transformed and expressed as colony-forming units per gram of fresh weight of forage. The log-transformed microbial data before ensiling and the chemical composition data (before ensiling and after 60 days of fermentation) were submitted to a one-way analysis of variance (ANOVA) and polynomial contrast to examine the effects of the unequally spaced altitude. Fermentation quality data were subjected to two-way ANOVA for a variance of 3 (sites) $\times$ 4 (ensiling times) factorial arrangement by Statistical Package for the Social Sciences (SPSS version 27.0, SPSS, Inc., Chicago, IL). Duncan's honestly significant difference (HSD) test was used for pair-wise comparisons, and the significance was reported at $P < 0.05$.

## ACKNOWLEDGMENTS

This research was supported by the joint key project of the National Natural Science Foundation of China (U20A2002).

## AUTHOR AFFILIATIONS

[1]State Key Laboratory of Grassland Agro-ecosystems, Lanzhou University, Lanzhou, China
[2]Probiotics and Biological Feed Research Centre, School of Life Sciences, Lanzhou University, Lanzhou, China
[3]College of Grassland Science, Gansu Agricultural University, Lanzhou, China
[4]State Key Laboratory of Plateau Ecology and Agriculture, Key Laboratory of Plateau Grazing Animal Nutrition and Feed Science of Qinghai Province, Academy of Science and Veterinary Medicine of Qinghai University, Xining, China
[5]Department of Animal Science, Ningxia University, Yinchuan, China

## AUTHOR ORCIDs

Fuhou Li  http://orcid.org/0000-0003-1070-8478
Xusheng Guo  http://orcid.org/0000-0002-5587-3920

## FUNDING

| Funder | Grant(s) | Author(s) |
| --- | --- | --- |
| MOST | National Natural Science Foundation of China (NSFC) | U20A2002 | Xusheng Guo |

## DATA AVAILABILITY

The sequenced metadata have been deposited in NCBI's Sequence Read Archive (SRA) under BioProject accession no. PRJNA1027119.

## ADDITIONAL FILES

The following material is available online.

### Open Peer Review

**PEER REVIEW HISTORY (review-history.pdf).** An accounting of the reviewer comments and feedback.

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
