## [Reviewer comments · Microbiology Spectrum]

Microbiology Spectrum

Responses of microbial community dynamics, co-occurrences, functional shifts, and natural fermentation profiles of *Elymus nutans* silage to altitudinal gradients

Rina Su, Fuhou Li, Ying Liang, Neha Sheoran, Jie Bai, Lizhuang Hao, Wencan Ke, Chen Hu, Mengya Jia, Samaila Usman, Mengyan Chen, and Xusheng Guo

Corresponding Author(s): Xusheng Guo, Lanzhou University

Review Timeline:

Submission Date:	June 15, 2023
Editorial Decision:	September 25, 2023
Revision Received:	October 25, 2023
Accepted:	November 7, 2023

Editor: Luca Cocolin

Reviewer(s): The reviewers have opted to remain anonymous.

Transaction Report:

DOI: <https://doi.org/10.1128/spectrum.02516-23>

September 25, 2023

Prof. Xusheng Guo
Lanzhou University
No.222 Tianshui South Road
Lanzhou, Gansu
China

Re: Spectrum02516-23 (Responses of microbial community dynamics, co-occurrences, functional shifts, and natural fermentation profiles of *Elymus nutans* silage to altitudinal gradients)

Dear Prof. Xusheng Guo:

Link Not Available

Sincerely,

Luca Cocolin

Journals Department
Reviewer comments:

Reviewer #1 (Comments for the Author):

This manuscript aligns well with the journal's scope, presenting intriguing findings with practical implications. To elaborate, the manuscript's conformity with the journal's focus is commendable, as it provides valuable insights that are in line with the journal's objectives. Nevertheless, there are certain issues that necessitate resolution prior to its acceptance.

1. There are some grammatical errors that need to be checked carefully for the whole manuscript.
2. Line 82-85: The sentence is somewhat lengthy and contains multiple ideas. To enhance clarity and readability, it might be helpful to break it down into two sentences. This would allow each idea to be conveyed more succinctly.

3. Line 102-104: If there are any prior studies or references that support this hypothesis, it would be beneficial to cite them to demonstrate that the hypothesis is grounded in existing scientific knowledge.
4. Line 156-158: The sentence is clear and concise in presenting the LA/AA ratio trends and regional differences. However, it might be helpful to provide specific numerical values or ranges for LA/AA ratios to offer readers a more precise understanding of the data.
5. Mention full form instead of just using abbreviation of PCoA and LEfSc.
6. Line 173-174: Consider briefly explaining the significance of these dominant genera in the context of silage fermentation or quality. Do *Pseudomonas* and *Xanthomonas* have specific roles or implications for silage preservation or other processes? Providing such context can help readers understand the relevance of these findings.
7. Line 239: Consider providing a brief explanation of why these specific KEGG pathways are relevant or important in the context of silage fermentation. This would help readers understand the implications of the findings and the significance of these pathways.
8. Line 258: Ensure that scientific terminology and abbreviations are consistently used and defined or explained where necessary to ensure accessibility to a broad readership.
9. Line 447: It's important to emphasize that the methods used were selected for their scientific validity and appropriateness for the study's objectives. If there were any modifications or deviations from the methodology in the report by Guo et al., these should be mentioned to ensure transparency.
10. Line 359: While the sentence is generally clear, it might be beneficial to avoid the phrase "As the case may be" for greater clarity. Instead, you can directly state the explanation without conditional phrasing.
11. Line 456: As for grammar and style, the sentence is well-structured, but the use of passive voice in phrases like "was applied," "were calculated," and "was conducted" can be somewhat passive and might benefit from more active language for added engagement and clarity.
12. Line 651-656: The legend of Fig. 3 is not very readable, please rewrite it.
13. Have to check all references and provide a uniform format.

Reviewer #2 (Comments for the Author):

This study investigated the changes in natural fermentation profiles, microbial communities, and their functional shifts in *Elymus nutans* silage with the altitudinal gradients. The topic is of interest and fits the scope of *Microbiology Spectrum*, but there are still many problems to be resolved. Please see the further and more detailed comments below.

1. The "Abstract" is an important part of the article, please write more simpler and clearly.
2. Suggest finding a native speaker to revise some grammar/language issues of the whole manuscript, such as Lines 26-38, 123-125, 177-180, and 260-264. Please reorganize these sentences and improve their readability.
3. Placed the abbreviations at the end of the manuscript, have to ensure the abbreviation in the "result" appeared the first time.
4. Line 169-170: The proportion of the "others" in Fig. 2 is very high, please explain the reason.
5. Line 210-221: This part is a bit like a discussion, there still lacks many descriptions about the results of the bacterial cooccurrence, complexity, and stability. Please add some results.
6. Line 273-275: There are some grammar problems. This sentence is comprised of two different subjects, suggesting to revise it as "Silage fermentation is an epiphytic microorganism-driven process, which leads to organic acid accumulation and a decrease in pH."
7. Line 298-300: it is not included in the report of the Ding et al (14) regarding of WSC and ADF contents, please check again.
8. Line 411-414: The influence of the altitude has been discussed in the "Discussion". However, I still hesitated that there are many factors including temperatures, soil fertilization, and rain fall of these sites also different among these three sites. So please add some descriptions about other environmental factors.

Staff Comments:

Preparing Revision Guidelines

- Point-by-point responses to the issues raised by the reviewers in a file named "Response to Reviewers," NOT IN YOUR COVER LETTER.
- Upload a compare copy of the manuscript (without figures) as a "Marked-Up Manuscript" file.
- Each figure must be uploaded as a separate file, and any multipanel figures must be assembled into one file.
- Manuscript: A .DOC version of the revised manuscript

- Figures: Editable, high-resolution, individual figure files are required at revision, TIFF or EPS files are preferred

Please return the manuscript within 60 days; if you cannot complete the modification within this time period, please contact me. If you do not wish to modify the manuscript and prefer to submit it to another journal, please notify me of your decision immediately so that the manuscript may be formally withdrawn from consideration by Microbiology Spectrum.

This study investigated the changes in natural fermentation profiles, microbial communities, and their functional shifts in *Elymus nutans* silage with the altitudinal gradients. The topic is of interest and fits the scope of Microbiology Spectrum, but there are still many problems to be resolved. Please see the further and more detailed comments below.

1. The “Abstract” is an important part of the article, please write more simpler and clearly.
2. Suggest finding a native speaker to revise some grammar/language issues of the whole manuscript, such as Lines 26-38, 123-125, 177-180, and 260-264. Please reorganize these sentences and improve their readability.
3. Placed the abbreviations at the end of the manuscript, have to ensure the abbreviation in the “result” appeared the first time.
4. Line 169-170: The proportion of the “others” in Fig. 2 is very high, please explain the reason.
5. Line 210-221: This part is a bit like a discussion, there still lacks many descriptions about the results of the bacterial cooccurrence, complexity, and stability. Please add some results.
6. Line 273-275: There are some grammar problems. This sentence is comprised of two different subjects, suggesting to revise it as “Silage fermentation is an epiphytic microorganism-driven process, which leads to organic acid accumulation and a decrease in pH.”.
7. Line 298-300: it is not included in the report of the Ding et al (14) regarding of WSC and ADF contents, please check again.

8. Line 411-414: The influence of the altitude has been discussed in the “Discussion”.

However, I still hesitated that there are many factors including temperatures, soil fertilization, and rain fall of these sites also different among these three sites. So please add some descriptions about other environmental factors.

Responses to the reviewer's and editor's comments

Introductory remark

We have carefully revised our manuscript (Spectrum02516-23) based on the comments and suggestions kindly provided by the reviewers and editors. The comments and corrections suggested by the referees were completely adopted unless the corresponding text had been otherwise modified. In the revised version prepared, all changes made in the text are colored. Texts highlighted in purple are the responses to Reviewer #1; the blue-colored texts are the responses to the comments of Reviewer #2; the red-colored texts are language and grammar modifications. We highly appreciate the constructive and kind comments that all the reviewers addressed on our manuscript.

Comments from the Editors:

Response: Raw data sequencing files and the associated metadata have been deposited in NCBI's Sequence Read Archive (PRJNA1027119).

Comments from the Reviewers:

Reviewer #1: This manuscript aligns well with the journal's scope, presenting intriguing findings with practical implications. To elaborate, the manuscript's conformity with the journal's focus is commendable, as it provides valuable insights that are in line with the journal's objectives. Nevertheless, there are certain issues that necessitate resolution prior to its acceptance.

1. There are some grammatical errors that need to be checked carefully for the whole manuscript.

Response: Thanks for your suggestions. A native speaker has reviewed and revised all the grammar issues in the entire manuscript. Please see the red-marked texts.

2. Line 82-85: The sentence is somewhat lengthy and contains multiple ideas. To enhance clarity and readability, it might be helpful to break it down into two sentences. This would allow each idea to be conveyed more succinctly.

Response: We have divided this long sentence into more readable two sentences, please see lines 81-84.

3. Line 102-104: If there are any prior studies or references that support this hypothesis, it would be beneficial to cite them to demonstrate that the hypothesis is grounded in existing scientific knowledge.

Response: We have cited the associated references, please see line 103.

4. Line 156-158: The sentence is clear and concise in presenting the LA/AA ratio

trends and regional differences. However, it might be helpful to provide specific numerical values or ranges for LA/AA ratios to offer readers a more precise understanding of the data.

Response: We have added the specific numerical ranges for LA/AA ratios in HY, XH, and CD silage samples during the entire fermentation process. Please see lines 155-157.

5. Mention full form instead of just using abbreviation of PCoA and LEfSc.

Response: The full form of PCoA and LEfSe have been mentioned in Line 162 and Line 199, respectively.

6. Line 173-174: Consider briefly explaining the significance of these dominant genera in the context of silage fermentation or quality. Do *Pseudomonas* and *Xanthomonas* have specific roles or implications for silage preservation or other processes? Providing such context can help readers understand the relevance of these findings.

Response: We have provided the contexts about the distribution and potential role of *Pseudomonas* and *Xanthomonas* in agriculture, which may be helpful in better understanding of findings. Please refer to the lines 354-360.

7. Line 239: Consider providing a brief explanation of why these specific KEGG pathways are relevant or important in the context of silage fermentation. This would help readers understand the implications of the findings and the significance of these pathways.

Response: We have added a brief explanation about the relevance or importance of

these specific KEGG pathways to silage fermentation process. Please see the lines 251-257.

8. Line 258: Ensure that scientific terminology and abbreviations are consistently used and defined or explained where necessary to ensure accessibility to a broad readership.

Response: The full form of UV is first mentioned, please refer to line 86.

9. Line 447: It's important to emphasize that the methods used were selected for their scientific validity and appropriateness for the study's objectives. If there were any modifications or deviations from the methodology in the report by Guo et al., these should be mentioned to ensure transparency.

Response: We have revised the description about Pacbio SMRT sequencing and raw data processing according to the methodology of Li et al. (36), please see lines 475-484.

Reference

36. Li X, Chen F, Wang X, Sun L, Guo L, Xiong Y, Wang Y, Zhou H, Jia S, Yang F, Ni K. 2021. Impacts of low temperature and ensiling period on the bacterial community of Oat silage by SMRT. *Microorganisms* 9:1–13.

10. Line 359: While the sentence is generally clear, it might be beneficial to avoid the phrase "As the case may be" for greater clarity. Instead, you can directly state the explanation without conditional phrasing.

Response: We have removed “As the case may be” and revised the sentence, please refer to lines 385-389.

11. Line 456: As for grammar and style, the sentence is well-structured, but the use of passive voice in phrases like "was applied," "were calculated," and "was conducted" can be somewhat passive and might benefit from more active language for added engagement and clarity.

Response: Thank you for your suggestion. We've revised the sentence as your recommendations. Please refer to lines 487-494.

12. Line 651-656: The legend of Fig. 3 is not very readable, please rewrite it.

Response: We have rewritten the legend of Fig 3, please refer to lines 712-718.

13. Have to check all references and provide a uniform format.

Response: We have checked carefully all references.

Reviewer #2 (Comments for the Author):

This study investigated the changes in natural fermentation profiles, microbial communities, and their functional shifts in *Elymus nutans* silage with the altitudinal gradients. The topic is of interest and fits the scope of Microbiology Spectrum, but there are still many problems to be resolved. Please see the further and more detailed comments below.

1. The "Abstract" is an important part of the article, please write more simpler and clearly.

Response: We have rewritten the "Abstract", please see lines 24-44.

2. Suggest finding a native speaker to revise some grammar/language issues of the whole manuscript, such as Lines 26-38, 123-125, 177-180, and 260-264. Please reorganize these sentences and improve their readability.

Response: A native speaker revised all these grammar/language issues thoroughly.

Please see the blue-colored texts such as lines 25-36, 124-125, 176-180, and 286-291.

3. Placed the abbreviations at the end of the manuscript, have to ensure the abbreviation in the "result" appeared the first time.

Response: Full forms of all abbreviations have been mentioned when first appeared in the “Introduction” or “Result”.

4. Line 169-170: The proportion of the "others" in Fig. 2 is very high, please explain the reason.

Response: We selected the top 30 bacterial taxa to show the bacterial community structure of the forage samples, while the remaining taxa have been classified into “others”. Understandably, well-preserved silage is indicated by low pH (preferably to < 4.0) and low alpha diversity, and is mainly dominated by the acid-tolerant *Lactiplantibacillus*. However, in this study, after 60 days of fermentation, the pH of all silage groups was more than 4.40. Moreover, the microbial community structures of these ensiled forages were highly complex and *Lactiplantibacillus* accounted for merely less than 30%. This can explain the high proportion of “others” in Fig. 2.

5. Line 210-221: This part is a bit like a discussion, there still lacks many descriptions about the results of the bacterial cooccurrence, complexity, and stability. Please add some results.

Response: We have revised this part and added some descriptions about the results, please see the lines 220-232.

6. Line 273-275: There are some grammar problems. This sentence is comprised

of two different subjects, suggesting to revise it as "Silage fermentation is an epiphytic microorganism-driven process, which leads to organic acid accumulation and a decrease in pH."

Response: Thank you for your suggestion. We have revised the sentence as your recommendations, please refer to lines 299-301.

7. Line 298-300: it is not included in the report of the Ding et al (14) regarding of WSC and ADF contents, please check again.

Response: We checked the description carefully and revised. Please refer to lines 317-319.

8. Line 411-414: The influence of the altitude has been discussed in the "Discussion". However, I still hesitated that there are many factors including temperatures, soil fertilization, and rain fall of these sites also different among these three sites. So please add some descriptions about other environmental factors.

Response: Descriptions about "other environmental factors" have been added, please see lines 275-284.

Re: Spectrum02516-23R1 (Responses of microbial community dynamics, co-occurrences, functional shifts, and natural fermentation profiles of *Elymus nutans* silage to altitudinal gradients)

Dear Prof. Xusheng Guo:

Your manuscript has been accepted, and I am forwarding it to the ASM production staff for publication. Your paper will first be checked to make sure all elements meet the technical requirements. ASM staff will contact you if anything needs to be revised before copyediting and production can begin. Otherwise, you will be notified when your proofs are ready to be viewed.

Sincerely,
Luca Cocolin
Editor
Microbiology Spectrum

Reviewer #2 (Comments for the Author):

This manuscript has been greatly improved based on the feedback from the review, and it is recommended for acceptance.